# Investigating the Chemical Composition of *Lepidium sativum* Seeds and Their Ability to Safeguard against Monosodium Glutamate-Induced Hepatic Dysfunction

**DOI:** 10.3390/foods12224129

**Published:** 2023-11-15

**Authors:** Manal Salah El-Gendy, Eman Sobhy El-Gezawy, Ahmed A. Saleh, Rashed A. Alhotan, Mohammed A. A. Al-Badwi, Elsayed Osman Sewlim Hussein, Hossam M. El-Tahan, In Ho Kim, Sungbo Cho, Sara Mahmoud Omar

**Affiliations:** 1Nutrition and Food Science Department, Faculty of Home Economics, Al-Azhar University, Tanta 31732, Egypt; 2Department of Poultry Production, Faculty of Agriculture, Kafrelsheikh University, Kafrelsheikh 333516, Egypt; 3Department of Animal Production, College of Food & Agriculture Sciences, King Saud University, P.O. Box 2460, Riyadh 11451, Saudi Arabia; 4AlKhumasia for Feed and Animal Products, Riyadh-Olaya-Al Aqareyah 2-Office 705, P.O. Box 8344, Riyadh 11982, Saudi Arabia; 5Animal Production Research Institute, Agricultural Research Center, Ministry of Agriculture, Dokki 12611, Egypt; 6Animal Resource and Science Department, Dankook University, Cheonan 31116, Republic of Korea; 7Smart Animal Bio Institute, Dankook University, Cheonan 330714, Republic of Korea

**Keywords:** monosodium glutamate (MSG), *Lepidium sativum* seeds (LSS), hepatotoxicity, antioxidant, dietary study, glutathione reductase

## Abstract

Monosodium glutamate (MSG) is one of the most frequently used food additives that endanger public health. The antioxidant, hyperlipidemic, and cytoprotective properties of *Lepidium sativum* seeds (LSS) as a natural remedy can minimize the harmful effects of MSG. This study investigated the potential protective effect of LSS against MSG-induced hepatotoxicity in rats. Male albino Sprague Dawley rats (*n* = 24) were equally divided into four groups for 30 days: the control group (G1) received a basal diet without supplement, group (G2) was fed a basal diet + MSG (30 g/kg b.w.) as a model group, group (G3) was fed a basal diet + MSG (30 g/kg b.w.) + LSS (30 g/kg b.w.), and group (G4) was fed a basal diet + MSG (30 g/kg b.w.) + LSS (60 g/kg b.w.). LSS enhanced serum alkaline phosphatase activity as well as total cholesterol, triglyceride, and glucose levels. It can decrease peroxide content in serum lipids and inhibit glutathione reductase and superoxide dismutase in hepatic cells. The dietary supplementation with LSS provided cytoprotection by enhancing the histoarchitecture of the liver and decreasing the number of apoptotic cells. Due to their antioxidant and anti-apoptotic properties, LSS effectively protect against the hepatotoxicity of MSG. These findings are of the highest significance for drawing attention to incorporating LSS in our food industry and as a health treatment in traditional medicine to combat MSG-induced hepatic abnormalities.

## 1. Introduction

The liver, the largest gland in the mammalian body, is essential for numerous metabolic processes. Hepatocytes, the primary cells of the liver, are responsible for vital functions such as detoxification, gluconeogenesis, glycogen storage, conversion of carbohydrates and proteins into lipids, synthesis of lipoproteins, phospholipids, and cholesterol, oxidation of fatty acids, storage of ferritin as a form of iron, and storage of vitamins A, D, and B12; however, liver function tests must be done frequently to determine the health of the liver [1]. The liver can self-regenerate; only 10 to 20 percent of a functioning liver is required for survival. It performs over 500 tasks, including the metabolism of nutrients, vitamin and mineral storage, bile excretion, and toxin detoxification [2]. The liver processes carbohydrates and stores glucose as glycogen, releasing it when needed. Several diseases, such as viral hepatitis, nonalcoholic fatty liver disease, alcoholic liver disease, and genetic disorders, can influence the role of the liver [3].

Food additives are increasingly used to enhance the quality of food. They can have immediate or long-term harmful effects on health, including headaches, changes in energy levels, and immune responses. MSG is a familiar flavor enhancer used as a food additive [4]. The taste of MSG is mediated by glutamate receptors (T1R1 + T1R3) in the taste buds and stomach [5]. Glutamate receptors (T1R1/T1R3 and mGluR1) are located throughout the gastrointestinal tract, and activation of these receptors has been proposed to affect nutrient absorption through the regulation of satiety hormones, including cholecystokinin (CCK) [6]. In addition to increasing glutamate levels in the blood plasma, it has been linked to brain disorders, oxidative stress, damaged livers, and renal toxicity [7].

There are three types of metabotropic glutamate receptors (mGluR) and four types of ionotropic glutamate receptors (NMDA, AMPA, delta, and kainite receptors). Various receptor types are present throughout the central nervous system, particularly the hypothalamus, hippocampus, and amygdala, which regulate autonomic and metabolic functions [8].

In recent years, the safety and toxicity of MSG have become controversial due to reports of adverse responses in individuals who have consumed MSG-containing foods. Numerous studies have confirmed the harmful effects of MSG [9]. MSG has been linked to headaches, nausea, diarrhea, irritable bowel syndrome, asthma symptoms in asthmatic patients, and panic attacks [10]. Moreover, Augustine et al. [10] concluded that MSG harms human health due to its association with Chinese Restaurant Syndrome (CRS). Long-term consumption of MSG can result in symptoms such as hepatotoxicity, renal injury, fibroids, and obesity.

Animal studies have provided a potential link between MSG and obesity. In these studies, neonatal administration of MSG has been used as a model to simulate obesity with impaired glucose tolerance and insulin resistance [11]. This has raised concerns about the impact of MSG consumption on human obesity. Several hypotheses have been proposed to explain how MSG may influence metabolism and contribute to obesity. One theory suggested that MSG could affect energy balance by increasing the palatability of food and disrupting the signaling cascade of leptin action in the hypothalamus [12].

Multiple studies have demonstrated the links between MSG-induced obesity and other systems. Injections of MSG during the first seven days of a rat’s existence led to MSG-induced obesity, which raised the mean arterial blood pressure and decreased heart rate variability, bradycardic responses, and vagal and sympathetic effects at 33 weeks relative to control rats [13]. The MSG-induced obesity model in mice demonstrated a significantly greater decrease in core temperature after acute exposure to cold (4 degrees Celsius for 2 h) and failed to mobilize brown adipose tissue lipids after exposure to 4 degrees Celsius for 6 h, whereas control animals did [14]. Ceglarek et al. [15] hypothesized that defective activation of thermogenic mechanisms in brown adipose tissue was responsible for MSG-treated mice’s defective cold-induced thermogenesis. Additionally, MSG can increase the expression of specific genes involved in inflammation, such as interleukin-6, tumor necrosis factor-alpha, resistin, and leptin, in visceral adipose tissue [16]. It has also been found to elevate insulin, resistin, and leptin levels in the bloodstream and impair glucose tolerance as MSG stimulates orosensory receptors and enhances the palatability of meals, which may contribute to weight gain [17].

MSG has been implicated in reducing the secretion of growth hormones, potentially leading to stunted growth and obesity. Excessive weight gain, primarily driven by the accumulation of excess fats in adipose tissue, can result in high cholesterol levels, cardiovascular diseases, liver disease, and endocrinological disorders [18].

*Lepidium sativum*, or garden cress, is a fast-growing herb with a peppery flavor and aroma. It is used medicinally to treat various conditions and possesses antipyretic, analgesic, and anti-inflammatory properties [19]. Its leaves contain sulforaphane and flavonol compounds, contributing to its antioxidant and anti-inflammatory effects. Lepidium *sativum* seeds are used as a remedy for inflammatory diseases such as diabetes, arthritis, traumatic injuries, and hepatitis in traditional medicine [20].

*Lepidium sativum* seeds (LSS) are reported to have various in vitro biological effects, including antioxidant, anti-inflammatory, antidiarrheal, antimicrobial, antispasmodic, and hepatoprotective action against oxidative damage, and have an excellent potential for use as herbal hepatoprotective or dietary supplements [21].

This study aimed to investigate the effect of LSS against hepatic effects induced by MSG in experimental rats.

## 2. Materials and Methods

This study followed regulations established by Kafrelsheikh University in Egypt (Number 4/2016 EC) and approved by the local experimental animal care ethics committee.

### 2.1. Materials

−*Lepidium sativum* seed powder was purchased from Imtenan Health Shop Company, Obour City, Egypt.−Casein (85%), cellulose, choline chloride, Maltodextrin, L-cysteine, minerals, and vitamins have been purchased from Al-Gomhoria Company for Chemicals, Cairo, Egypt.−Serum and Vaccine Center in Cairo, Egypt, provided 24 healthy male albino rats weighing (150 ± 10 g) of the Sprague Dawley strain. The rats were five weeks old.

### 2.2. Methods

#### 2.2.1. Biological Experiment

##### Animals

A total of 24 rats were housed in individual wire cages with wire bottoms, ensuring hygienic conditions. Specialized feeding containers were employed in order to avoid the dispersion of the rats’ food. Furthermore, the water supply was provided ad libitum and subjected to daily examination.

##### Diets

The diet consisted of high-quality ingredients per 100 g. Based on the study’s findings of Arriarán et al. [22], the meal composition consisted of 10% sunflower oil, 4% salt mixture, 1% vitamin mixture, 0.3% DL-methionine, 0.2% choline chloride, and 14% protein. Notably, the inclusion of protein was adjusted by substituting up to 100 g of corn starch. The diet included Lepidium sativum seeds as whole, grounded, dry seeds.

#### 2.2.2. Experimental Design

Rats were housed in clean, well-ventilated cages and fed a basal diet for three days to facilitate adaptation. The rats were then divided into the following four groups:−G1 (−Ve) group: the negative control group was fed a basal diet;−G2 (+Ve) group: the positive control group was fed a basal diet + MSG (30 g/kg b.w.);−G3: fed a basal diet + MSG (30 g/kg b.w.) + LSS powder (30 g/kg b.w.);−G4: fed a basal diet + MSG (30 g/kg b.w.) + LSS powder (60 g/kg b.w.).

The dose of LSS powder (30 and 60 g/kg b.w.) was applied according to Abd-Elkareem et al. [23].

During the 30-day experiment, daily records were kept of the amount of diet consumed or wasted, as well as the rats’ weekly weight. The rats were fasted overnight and sacrificed at the end of the experiment, and blood samples were collected and centrifuged at 3000 rpm for 10 min to obtain serum. For analysis, the serum was transferred to dry, sterile Eppendorf tubes and frozen at −20 °C. Each rat’s liver was extracted, washed with a saline solution, weighed, and submerged in a formalin solution. The specimens were trimmed, dehydrated, sectioned, and stained with hematoxylin and eosin, as Carleton [24] described.

#### 2.2.3. Biological Evaluation

Upon completion of the experiment, many parameters were measured to assess the outcomes. These included feed intake (FI), body weight gain percentage (BWG%), organ weight as a percentage of total body weight (ROW%), and feed efficiency ratio (FER).

#### 2.2.4. Determination of Total Phenolic Contents in Lepidium Sativum Seeds

Using the Agilent 1260 series, HPLC analysis was conducted. The separation was accomplished using an Eclipse C18 column (4.6 mm × 250 mm; 5 m). At a flow rate of 0.9 mL/min, the mobile phase consisted of water (A) and 0.05% trifluoroacetic acid in acetonitrile (B) in a mixture. The stage of mobility was consecutively programmed with the following linear gradient: 0 min (82%A); 0–5 min (80%A); 5–8 min (60%A); 8–12 min (60%A); 12–15 min (82%A); 15–16 min (82%A); and 16–20 min (82%A). The multi-wavelength detector was observed at 280 nm. The injection volume for each sample solution was 5 mL. The temperature of the column was maintained at 40 °C.

#### 2.2.5. Chemical Analysis

The moisture, protein, fat, fiber, and ash contents of *Lepidium sativum* seeds were analyzed using the procedures outlined in the AOAC [25] guidelines. The method of difference determined the estimation of carbohydrate content.

The determination of mineral contents was conducted using the AOAC [26] technique.

#### 2.2.6. Biochemical Analysis of Serum

##### Lipid Profile

Specialized tests and a spectrophotometer (model T80, UV/visible, double beam, UK) were used to measure the amounts of total cholesterol (TC) [27], triglycerides (TG) [28], and high-density lipoprotein cholesterol (HDL-c) [28]. Murakami et al. [29] stated that the calculation of low-density lipoprotein cholesterol (LDL-c) was performed.

##### Liver Enzymes

The determination of serum levels of aspartate aminotransferase (AST), alanine aminotransferase (ALT), and alkaline phosphatase (ALP) was conducted in accordance with the methodology outlined by Sampson et al. [30]. The total protein and albumin were assessed in the study conducted by Nasrin and Alim [31]. The equation utilized to determine globulin was documented by Kazmi et al. [32].

Globulin = Total protein − Albumin

##### Oxidant/Antioxidant Activity in Liver Tissue

The liver tissue that had been homogenized into an aqueous solution was used to test the different amounts of antioxidants. Adaway et al. [33] recognized malondialdehyde (MDA) and nitric oxide (NO) as the resultant substances of lipid peroxidation. The colorimetric technique was used to evaluate the levels of endogenous antioxidant systems, such as superoxide dismutase (SOD) and catalase (CAT). Lipid peroxidation is the degradation of lipids in cell membranes, resulting from reactive oxygen species (ROS) attacking unsaturated fatty acids. This process produces lipid peroxides, which can further react with other molecules, generating additional reactive species. Malondialdehyde (MDA) is a byproduct used as a biomarker for oxidative stress. Nitric oxide (NO) is a byproduct of lipid peroxidation generated by reactive species and can impact cellular signaling and oxidative stress pathways [34].

#### 2.2.7. Histopathological Examinations

The livers of the sacrificed rats were immersed in a solution containing 10% formalin. The livers that had been preserved in formalin were subjected to a dehydration process using increasing concentrations of ethanol. Subsequently, they were cleaned using methyl benzoate and finally embedded in paraffin wax. The liver samples were prepared by cutting paraffin slices that were 4–6 μm thick. These sections were then subjected to histological staining using specific techniques. Haematoxylin and eosin stain were used for a general histological evaluation of the liver, while the Periodic Acid Schiff (PAS) technique was employed to see glycogen, following the method described by Sarker and Oba [35].

#### 2.2.8. Statistical Analysis

Data were expressed as the mean ± standard deviation (mean ± SD) for six rats in each group. Statistical differences between groups were identified by one-way analysis of variance (ANOVA), followed by the Duncan post-test. Data are the average of three replicates ± the standard deviation (%). All statistical analyses were done using SPSS for Windows software, version 16.0. (SPSS, Inc., Chicago, IL, USA). A probability (*p*) value of <0.05 was considered statistically significant.

## 3. Results and Discussion


**The composition of LSS powder g/100 g dry weight**


Table 1 shows the LSS powder chemical composition. The results indicate that the macronutrient content of LSS is high and consistent with previous studies. The constituents of these seeds are 25% protein, 14–24% lipids, 35–54% carbohydrates, and 8% crude fiber. According to Sonzogni et al. [36] and Attia et al. [37], LSS have a high nutritional value, with moisture, crude fat, protein, carbohydrate, fiber, and ash contents of 2.9%, 23.2%, 24.2%, 30.7%, 11.1%, and 7.1%, respectively.

Table 2 presents the LSS mineral composition, which contains a high level of potassium and phosphorus, as well as moderate levels of calcium, sodium, and magnesium. A value of 2.96 mg/100 g was discovered to be a low zinc level. These findings agree with Nasef and Khateib [38], who stated that LSS are an excellent source of minerals, including phosphorus, magnesium, and potassium.

The phenolic compounds of LSS are presented in Table 3. The highest phenolic compounds of LSS were observed for gallic and chlorogenic acid, catechin, and pyrocatechol. These findings suggest that phytochemicals such as gallic acid and protocatechuic acid in LSS may improve human health by contributing to the antioxidant defense system against free radicals. Our results are consistent with those El-Salam et al. [39] reported, who also stated the high antioxidant capacity of these phytochemicals. Additionally, research by Chatoui et al. [40] has shown that flavonoids, a type of phenolic compound found abundantly in plants, may have hypoglycemic and anti-diabetic properties.


**Impact of LSS powder on FI, BWG%, and FER in rats**


Results in Table 4 indicated that total feed intake for the control G1 (−Ve) was higher than the control G2 (+Ve). G3 and G4 groups were higher than the control G2 (+Ve), while G4 showed a highly significant difference (*p* < 0.001) compared to the control G2 (+Ve). The results revealed that the mean value of feed efficiency ratio (FER) and body weight gain percentage in the rats in group G2 (+Ve) was significantly lower relative to the G1 (−Ve) group. At the same time, all treated groups recorded a significant improvement compared to the control group G2 (+Ve). According to these results, hepatic dysfunction may be linked to reduced body weight, as Tavakoli et al. [41] reported. However, rats with hepatopathy who were given 30 or 60 g/kg b.w. of LSS showed an increase in body weight gain (BWG), feed intake (FI), and feed efficiency ratio (FER) versus positive control rats. These results are consistent with the findings of Hashemi and Alahmari [42], who observed a positive effect of LSS on appetite and nutrient digestion. Similarly, Ait-Yahia et al. [43] found that body weight gain decreased significantly in rats with induced hepatic dysfunction but increased with treatment using LSS.

In comparison to the negative group, the relative liver weight increased in the positive control group (G2). In contrast to the positive control (G2), all treated groups’ relative liver weight decreased significantly. Dietary supplementation with MSG (30 g/kg b.w.) and LSS (60 g/kg b.w.) produced the best results in the group (G4). While those who were fed a basal diet +MSG (30 g/kg b.w.) + *LSS* (30 g/kg b.w.) also showed a significant decrease compared with the positive control group (G2), as noticed in Figure 1. The increase in liver weight in the positive control group (G2) is consistent with the previous investigation by Doke and Guha [44], who also observed elevated liver weight, but this was reversed by administering LSS. Similarly, Akram et al. [45] reported that the LSS reduced liver fat and weight, possibly by reducing lipid accumulation in the body, leading to increased excretion of fat in feces, which was evident by the increase in fecal weight of animals fed with LSS versus the control group.


**Impact of dietary LSS powder on serum lipid profiles in rats**


The positive control group (G2) had a significantly higher mean value of TC and TG than the negative control group (G1), whereas the treatment groups had a significantly lower mean value of TC and TG than the positive control group (G2). Dietary supplementation with MSG (30 g/kg b.w.) and LSS (60 g/kg b.w.) produced the most outstanding results in the group (G4) (Figure 2).

Rats given MSG showed a significant rise in TC, TG, and LDL-c levels in their serum, as well as a reduction in HDL-c content. These results differ from those stated by Bárbara et al. [46], who noted that MSG increased the activity of coenzyme A (HMG CoA) reductase, 3-hydroxyl-3-methylglutaryl, and increased the synthesis and hyperlipidemia of cholesterol. This shift in glucose metabolism toward lipogenesis was also observed by Ibegbulem et al. [47].

Figure 3 shows the impact of LSS on HDL-c, LDL-c, and VLDL-c. The results show that HDL-c for control G2 (+Ve) was the lowest value; moreover, G4 led to a highly significant (*p* < 0.01) value compared to control G2 (+Ve). At the same time, control G2 (+Ve) had the highest LDL-c value, while control G2 (+Ve) had the highest VLDL-c among all the groups. In the current study, feeding rats LSS decreased triglycerides, LDL-c, and VLDL-c while increasing HDL-c. This finding is consistent with the findings of Diab and Hamza [48], who discovered that rats fed LSS had significantly decreased LDL-c and TG levels in the serum. The beneficial effects of LSS belong to the abundance of compounds with biological activity, such as flavonoids, alkaloids, and high levels of omega-3 fatty acids. These compounds have been shown to prevent cardiac attacks and strengthen the immune system [49].


**Impact of dietary LSS powder on hepatoprotective effects**


The mean values of AST, ALT, and ALP in the G2 (+Ve) control group increased significantly compared to the G1 (−Ve) control group (Figure 4). All treated groups demonstrated a significant decrease in hepatic enzyme levels relative to the control (+Ve) group. The best finding of AST, ALT, and ALP was found in the group (G4) that was fed a base diet plus MSG (30 g/kg b.w.) and LSS (60 g/kg b.w.).

The extent of hepatic damage is assessed by the increase in serum levels of the cytoplasmic enzymes AST, ALT, and ALP and by histopathological examination. The increased serum levels of AST and ALT have been attributed to damage to the structural integrity of the liver, as these cytoplasmic enzymes are released into circulation after cellular damage [50]. Elevation of AST has been reported to be an index of hepatocellular injury in rats, whereas ALT elevation is more commonly associated with the necrotic state, serum ALP, and GGT, which are essential enzymes for assessing obstructive liver injury [51].

MSG can cause liver damage and result in an increase in the liver enzymes ALT, AST, and ALP, according to its cytotoxic effects. MSG toxicity can also result in the formation of ammonium ions and reactive oxygen species (ROS), which can damage cell membranes and release hepatic enzymes [52]. MSG can induce oxidative stress; oxidative stress occurs when there is an imbalance between the production of free radicals, such as oxygen radicals and reactive oxygen species (ROS), and the body’s ability to eliminate them [53]. Although the exact mechanisms are not fully understood, growing evidence suggests that α-ketoglutarate dehydrogenase, glutamate receptors, and cysteine-glutamate antiporters are involved in the upregulation of oxidative stress in MSG-induced hepatotoxicity. Other factors, including nutrition, metabolism, hormones, cytokines, and detoxification processes, also contribute to oxidative stress [54].

Antioxidant compounds such as phenolic acids, polyphenols, and flavonoids can remove free radicals, including hydroxyl, hydroperoxide, and lipid peroxyl radicals, in addition to neutralizing hydrogen peroxide and superoxide anion, thereby inhibiting the oxidative harm that results in chronic illnesses [55]. *Lepidium sativum* seeds have been shown to possess antioxidant characteristics that can help prohibit inflammation, as demonstrated by Buso et al. [20].

There is also agreement between the present study and those by Nasab et al. [56]. In relation to the finding that rats subjected to MSG administration exhibited elevated serum ALT levels, this result can probably be attributed to MSG’s observed induction of hepatic oxidative stress [57].

The mean values of TP, Alb, and Glb in the G2 (+Ve) group were significantly higher than in the G1 (−Ve) group (Figure 5). At the same time, all treated groups showed a significant decrease relative to group G2 (+Ve), while the lowest levels of total protein, albumin, and globulin were determined in the group (G4) that received a diet containing MSG (30 g/kg b.w.) and LSS (60 g/kg b.w.). The present study found that feeding rats with LSS significantly increased serum total protein, albumin, and globulin levels in rats with monosodium glutamate-induced hepatotoxicity. Our results agree with an investigation by Sahin et al. [58], who found that LSS had a hepatoprotective effect by restoring serum enzyme levels to normal and increasing albumin and total protein. MSG impairs protein metabolism in injured hepatocytes, leading to a reduction in protein synthesis and an improvement in hepatic enzyme activities. LSS may restore TP and Alb levels by enhancing the poisoned liver’s functional status and protecting it from hepatotoxicity. Zamzami et al. [21] investigated the hepatoprotective effect of LSS in rabbits and found that it repaired liver injury markers and improved bilirubin, total protein, and albumin levels, further confirming its hepatoprotective effect.

The mean values of total bilirubin (T.BIL), direct bilirubin (direct BIL), and indirect bilirubin (indirect BIL) in the G2 (+Ve) group were significantly higher than all treated groups (Figure 6). T.BIL, direct BIL, and indirect BIL levels were the lowest in the group (G4). Treating rabbits with LSS significantly recovered their liver’s damaging marker enzymes and their T.BIL, protein, and albumin levels; this validates the hepatoprotective effect of LSS, as Okafor and Ezejindu [50] reported.


**Impact of Lepidium sativum seeds on antioxidant enzymes (CAT, SOD) and oxidative stress markers (NO, MDA) in rats**


Figure 7 shows the impact of LSS on antioxidants. CAT, SOD, and GPx activities in the liver tissue of group G4 rats were significantly decreased (*p* < 0.001). The (+Ve) G2 group had significantly higher lipid peroxide levels (MDA and NO) than the negative G1 group. Simultaneously, all treated groups demonstrated a significant reduction relative to group G2 (+Ve).

Oral administration of MSG was found to increase oxidative stress markers such as MDA, ROS, NO, and hydrogen peroxide (H_2_O_2_) while decreasing SOD, CAT, glutathione S transferase (GST), and reduced glutathione (GSH). Our results agree with the findings of [59], who stated that MSG consumption could lead to lipogenesis and depletion of NADH and GSH levels in the liver. The increased levels of MDA and NO may result from lipid peroxidation caused by an altered cell redox state due to difficulty in glutamate transportation across the cell membrane. However, the administration of LSS was found to normalize the oxidant status of liver cells and protect against free radicals and H_2_O_2_, in agreement with Zanfirescu et al.’s [60] findings on the antioxidant activity of LSS.


**Histopathology of Liver**


Bile canaliculi, small channels formed by adjacent hepatocytes, are visible between the hepatocyte plates. The portal triads are well-formed and comprise the hepatic artery, portal vein, and bile duct branches. No significant inflammation or fibrosis is noted in the portal tracts. The connective tissue framework, including the portal tracts and central vein region, appears intact and shows no evidence of fibrosis or abnormal collagen deposition. No significant infiltration of inflammatory cells, such as lymphocytes, plasma cells, or neutrophils, is observed within the hepatic lobules or portal tracts. The liver biopsy demonstrates a normal histological appearance. These findings are consistent with standard liver histopathology [H&E × 100] (Figure 8a).

The liver tissue demonstrates alterations in the normal architecture due to acute inflammation. The hepatic lobules exhibit disarray and distortion, with a loss of the typical hexagonal arrangement. The lobular organization is disrupted, and there is variation in hepatocyte size and shape. The hepatocytes show significant changes associated with acute inflammation. They appear swollen with cytoplasmic vacuolation, which can be attributed to the effect of monosodium glutamate (MSG). The nuclei of hepatocytes may exhibit hyperchromasia (increased nuclear staining) and pleomorphism (variations in size and shape).

The liver tissue displays prominent inflammatory cell infiltration, primarily lymphocytes and plasma cells. These cells are distributed diffusely within the hepatic lobules and portal areas. The inflammatory infiltrate indicates an immune response triggered by MSG; the sinusoids, which are vascular channels between the hepatocyte plates, demonstrate congestion and dilation, likely due to the inflammatory response. This congestion can impede the normal exchange of substances between the bloodstream and hepatocytes. The liver biopsy demonstrates histopathological features consistent with acute inflammation attributed to monosodium glutamate (MSG) intake. These features include alterations in liver architecture, hepatocyte changes with cytoplasmic vacuolation, inflammatory infiltrate, sinusoidal congestion, and portal triad inflammation. These findings support the diagnosis of acute liver inflammation associated with MSG exposure. [H&E × 100] (Figure 8b–d).

Figure 8e, which revealed that the liver tissue demonstrates changes in the normal architecture due to acute inflammation. The hepatic lobules exhibit disarray, with a loss of the typical hexagonal arrangement. The lobules may appear irregularly shaped and distorted. The hepatocytes show signs of acute inflammation, such as swelling and cytoplasmic vacuolation. However, administering Lepidium sativum seed powder, known for its anti-inflammatory properties, may mitigate hepatocyte alterations. The hepatocytes may appear less swollen and exhibit reduced cytoplasmic vacuolation compared to cases without Lepidium sativum seed intervention. The liver tissue displays an infiltrate of inflammatory cells, primarily consisting of lymphocytes and plasma cells. These cells are distributed within the hepatic lobules and portal areas. The presence of Lepidium sativum seed intervention may lead to a relatively decreased inflammatory infiltrate compared to cases without Lepidium sativum seed administration, suggesting a potential anti-inflammatory effect. The sinusoids, the vascular channels between the hepatocyte plates, may exhibit reduced congestion and dilation compared to cases without Lepidium sativum seed intervention. The anti-inflammatory properties of Lepidium sativum seeds may contribute to mitigating sinusoidal changes associated with acute inflammation. In cases of acute inflammation due to monosodium glutamate (MSG) with Lepidium sativum seed intervention, the development of fibrosis is generally not observed. The anti-inflammatory properties of Lepidium sativum seeds may help prevent or reduce fibrotic changes in liver tissue. The liver biopsy demonstrates histopathological features consistent with acute inflammation attributed to monosodium glutamate (MSG) intake. The administration of Lepidium sativum seeds, known for their anti-inflammatory properties, appears to have a mitigating effect on certain histopathological changes associated with acute liver inflammation, including hepatocyte alterations, inflammatory infiltrate, sinusoidal changes, and portal triad inflammation [H&E × 200].

Figure 8c, revealing portal infiltration with leucocytes and hydropic degeneration of hepatocytes, where a noticeable reduction in inflammatory responses is observed. The inflammatory cell infiltration in the portal tracts and parenchyma is diminished compared to the surrounding areas. The number of lymphocytes and neutrophils is significantly reduced, indicating a decline in the inflammatory process. The hepatocytes show less swelling and vacuolization compared to the surrounding areas. The cytoplasm appears more normalized, indicating a reduction in cellular damage and improved cellular function. Focal necrosis is observed to be resolving. The necrotic areas show signs of repair and regeneration, with infiltration of mononuclear cells and evidence of hepatocyte repopulation. This suggests a positive response to Lepidium sativum seed consumption in terms of tissue healing and restoration. [H&E × 200].

## 4. Conclusions

The present study illustrates that the seeds of *Lepidium sativum* powder benefit the hepatic dysfunction induced by monosodium glutamate. This was evidenced by improved body weight gain, reduced food and liquid intake, and enhanced liver function, as indicated by alterations in ALT, AST, ALP, and total bilirubin levels. Furthermore, the seeds positively affected serum albumin levels through the action of various essential antioxidants. Hence, it is plausible that the seeds of Lepidium sativum could potentially mitigate the adverse effects of monosodium glutamate, specifically hepatic impairment.

*Lepidium sativum*, often known as garden cress, is known to possess a diverse array of chemical constituents, such as fatty acids, proteins, shikimic acids, vitamins, carbohydrates, calcium, phosphorus, trace elements, and other compounds. Garden cress (*Lepidium sativum*) is commonly consumed as both a dietary staple and a medicinal resource. Its efficacy has been demonstrated in treating various ailments, including hypertension, arthritis, hepatotoxicity, inflammation, diabetes, cancer, bronchitis, and others. According to a comprehensive analysis of multiple studies, *Lepidium sativum* has demonstrated its inherent usefulness and merits further investigation into its potential nutritional and therapeutic applications.

## Figures and Tables

**Figure 1 foods-12-04129-f001:**
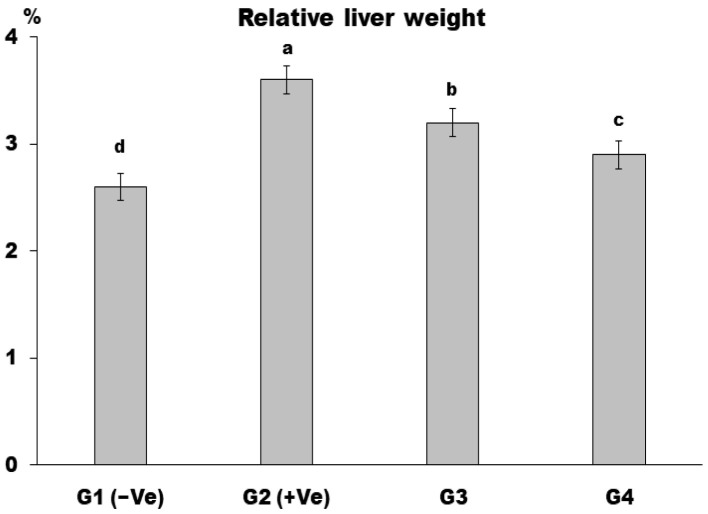
Relative liver weight % in control and differently treated rats. Data are presented as means ± SD for six rats in each group. The significant change is at *p* ≤ 0.05 (*n* = 6).

**Figure 2 foods-12-04129-f002:**
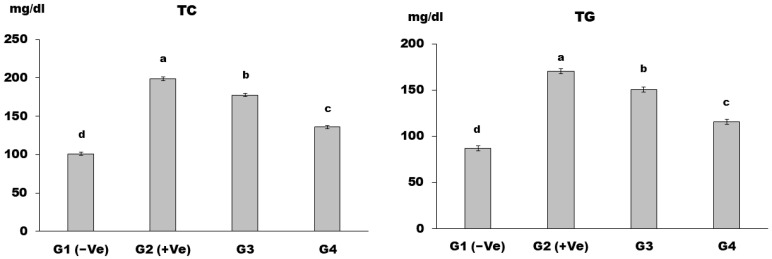
Total cholesterol (TC) and triglycerides (TG) in control and differently treated rats are shown in figures (a and b, respectively). Data are presented as the means ± SD for six rats in each group. The significant change is at *p* ≤ 0.05 (*n* = 6).

**Figure 3 foods-12-04129-f003:**
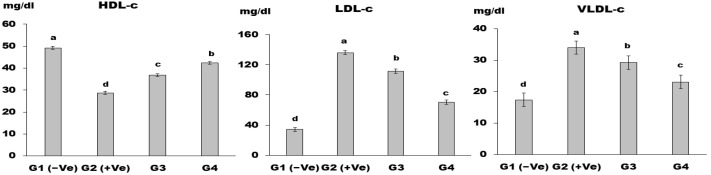
HDL-c, LDL-c, and VLDL-c in control and differently treated rats showed figures (a–c, respectively). Data are presented as means ± SD for six rats in each group. The significant change is at *p* ≤ 0.05 (*n* = 6).

**Figure 4 foods-12-04129-f004:**
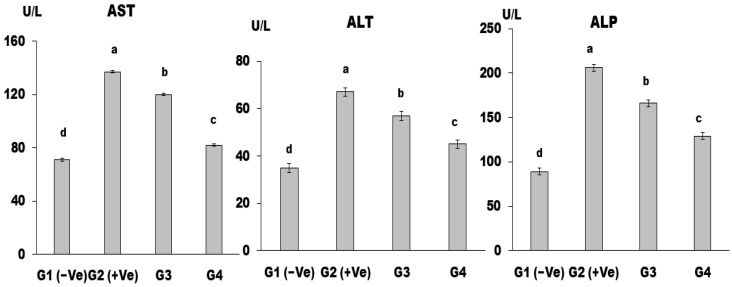
AST, ALT, and ALP in control and differently treated rats showed figures (a–c, respectively). Data are presented as means ± SD for six rats in each group. The significant change is at *p* ≤ 0.05 (*n* = 6).

**Figure 5 foods-12-04129-f005:**
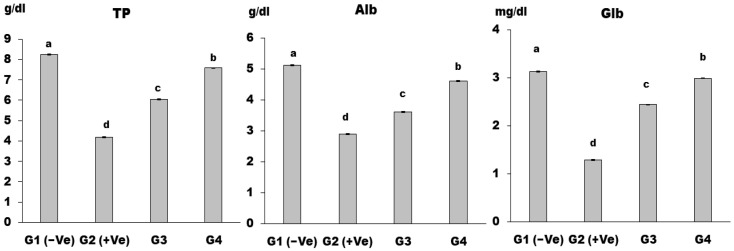
TP, Alb, and Glb in control and differently treated rats. Data are presented as means ± SD for six rats in each group. The significant change is at *p* ≤ 0.05 (*n* = 6).

**Figure 6 foods-12-04129-f006:**
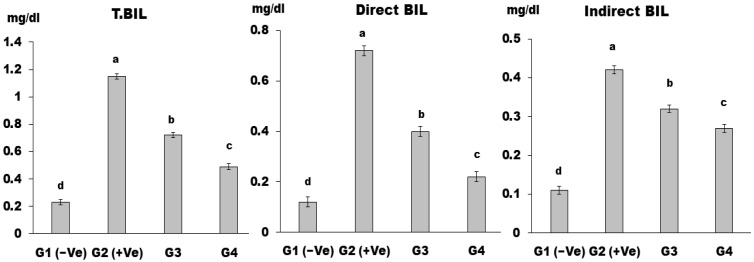
T.BIL, direct BIL, and indirect BIL in control and differently treated rats. Data are presented as means ± SD for six rats in each group. The significant change is at *p* ≤ 0.05 (*n* = 6).

**Figure 7 foods-12-04129-f007:**
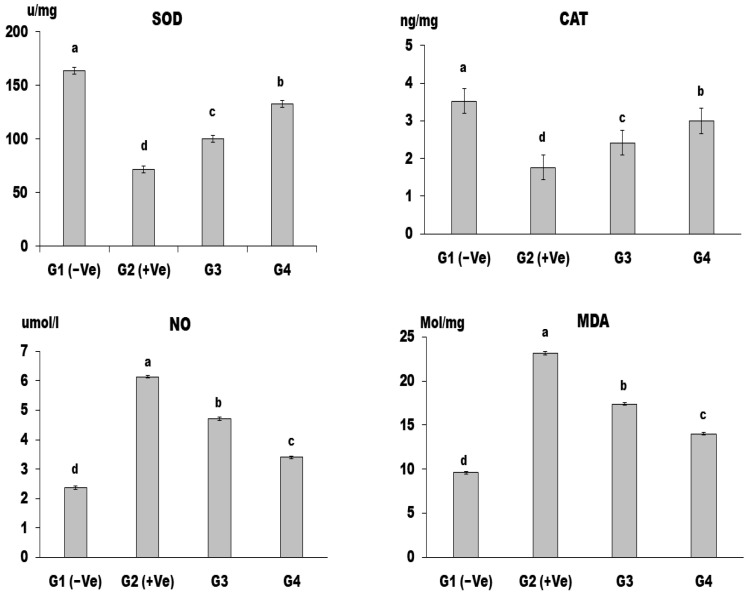
CAT, SOD, NO, and MDA in control and differently treated rats are shown in figures (a and b, respectively). Data are presented as means ± SD for six rats in each group. The significant change is at *p* ≤ 0.05 (*n* = 6).

**Figure 8 foods-12-04129-f008:**
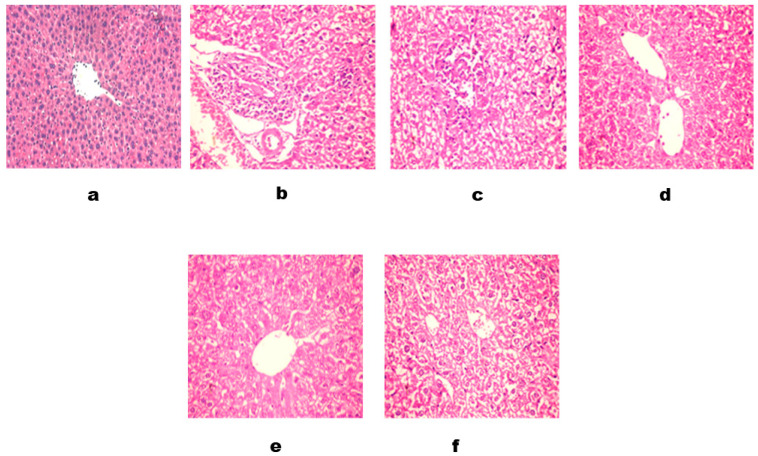
(**a**) (fed a basal diet) revealed Kupffer cells, specialized macrophages located within the sinusoids observed. (**b**) Section in the liver positive control group (G2) (fed a basal diet +MSG) showed focal hepatic necrosis associated with leucocytic cell infiltration. (**c**) portal infiltration with leucocytes and megakaryocytes. (**d**) portal infiltration with cholangitis. (**e**) The section in the liver of group (3) was fed a basal diet +MSG + LSS powder (30 g/kg b.w.). (**f**) The section in the liver of group (4) was fed a basal diet + MSG + LSS powder (60 g/kg b.w.).

**Table 1 foods-12-04129-t001:** Chemical composition of *Lepidium sativum* seed powder g/100 g dry weight.

Constituent of *Lepidium sativum* Seeds	g/100 g
Moisture	6.73 ± 0.02 ^cd^
Crude protein	21.61 ± 0.71 ^c^
Fat	32.28 ± 0.18 ^a^
Crude fiber	6.75 ± 0.06 ^cd^
Ash	4.83 ± 0.09 ^d^
Total carbohydrate	27.80 ± 0.73 ^b^

Data are the average of three replicates ± the standard deviation (%). Moisture, crude protein, fat, crude fiber, ash, and total carbohydrate. The results are expressed as mean ± SD. Values bearing dissimilar alphabets as superscripts vary significantly at *p* < 0.05 among the groups.

**Table 2 foods-12-04129-t002:** Mineral content of *Lepidium sativum* seed powder mg/100 g.

Minerals of LSS	mg/100 g
Potassium (K)	296.36 ± 10.23 ^ab^
Calcium (Ca)	210.23 ± 13.25 ^c^
Phosphorus (P)	944.33 ± 16.03 ^a^
Magnesium (Mg)	325 ± 3.64 ^b^
Sodium (Na)	230.35 ± 4.07 ^bc^
Zinc (Zn)	2.96 ± 0.15 ^d^

Data are the average of three replicates ± the standard deviation (%). The results are expressed as mean ± SD. Values bearing dissimilar alphabets as superscripts vary significantly at *p* < 0.05 among the groups.

**Table 3 foods-12-04129-t003:** Phenolic compounds in LSS powder.

Phenolic Compounds	Conc. (µg/g)
Gallic acid	111.86 ± 5.11 ^d^
Chlorogenic acid	238.16 ± 7.03 ^b^
Catechin	176.30 ± 4.01 ^c^
Coffeic acid	284.79 ± 11.06 ^ab^
Pyro catechol	408.41 ± 9.15 ^a^
Ellagic acid	75.60 ± 2.10 ^de^
Vanillin	4.91 ± 0.03
Ferulic acid	13.06 ± 0.09
Naringenin	19.94 ± 2.16
Quercetin	169.29 ± 4.17 ^c^
Cinnamic acid	18.27 ± 5.02
Hesperetin	51.87 ± 2.14 ^e^

Data are the average of three replicates ± the standard deviation (%). The results are expressed as mean ± SD. Values bearing dissimilar alphabets as superscripts vary significantly at *p* < 0.05 among the groups.

**Table 4 foods-12-04129-t004:** Impact of LSS powder on FI, BWG%, and FER in rats.

Groups	G1 (−Ve)	G2 (+Ve)	G3	G4
FI (g)	781.20 ± 2.27 ^a^	474.99 ± 2.42 ^d^	560.00 ± 1.50 ^c^	701.90 ± 1.15 ^b^
BWG%	50.40 ± 3.64 ^a^	32.40 ± 2.50 ^d^	38.40 ± 2.07 ^c^	42.60 ± 2.50 ^b^
FER	0.07 ± 0.005 ^a^	0.04 ± 0.002 ^d^	0.05 ± 0.003 ^c^	0.06 ± 0.004 ^b^

FI (feed intake), BWG (body weight gain), and FER (food efficiency ratio). G1 was the negative control group, G2 was the positive control group, G3 was fed a basal diet + MSG (30 g/kg b.w.) + LSS (30 g/kg b.w.), and G4 was fed a basal diet + MSG (30 g/kg b.w.) + LSS (60 g/kg b.w.). The results are expressed as mean ± SD for six rats in each group. Values bearing dissimilar alphabets as superscripts vary significantly at *p* < 0.05 among the groups.

## Data Availability

Data is contained within the article.

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
