# Peer review of "Investigating the Chemical Composition of Lepidium sativum Seeds and Their Ability to Safeguard against Monosodium Glutamate-Induced Hepatic Dysfunction"

_foods, 2023, doi:10.3390/foods12224129_

Round 1
Reviewer 1 Report
Comments and Suggestions for Authors
This study focused on the potential protective effect of LSS against MSG-induced hepatotoxicity, and it was investigated using rats. This paper presents significant data, but I think it is quite inadequate in the way the paper is written. I would like to point out the following points.
(1) MSG and LSS exposure levels are listed as 30g of diet, but I don't think they are accurately indicated. Shouldn't it be indicated by the amount of exposure per kg of animal weight?
(2) I think the exposure group settings are inadequate. I think they should have also considered a group exposed only to LSS.
Comments on the Quality of English Language
 I think the proposed presentation does not respect the elementary rules of a scientific writing. First, the citation method of the literature in the main text is not appropriate. Authors should not write with the document number as the subject. Also, I don't think the way the Figures are shown is correct. It is not shown exactly what the graph axes represent.
For these reasons, I cannot recommend accepting this paper. 
Author Response
Review Report Form 1
This study focused on the potential protective effect of LSS against MSG-induced hepatotoxicity, and it was investigated using rats. This paper presents significant data, but I think it is quite inadequate in the way the paper is written. I would like to point out the following points.
(1) MSG and LSS exposure levels are listed as 30g of diet, but I don’t think they are accurately indicated. Shouldn’t it be indicated by the amount of exposure per kg of animal weight?
Response: Thank you for your notice. We modified it to 30 or 60g/kg b.w.
(2) I think the exposure group settings are inadequate. I think they should have also considered a group exposed only to LSS.
Response: Thank you for your suggestion. A distinct group was not established for LSS powder in our investigation due to the focus on utilising the LSS powder for preventive rather than therapeutic purposes. Consequently, the rats were administered LSS powder concurrently with the induction substance.
I think the proposed presentation does not respect the elementary rules of a scientific writing. First, the citation method of the literature in the main text is not appropriate. Authors should not write with the document number as the subject. Also, I don’t think the way the Figures are shown is correct. It is not shown exactly what the graph axes represent. For these reasons, I cannot recommend accepting this paper. 
Response: Thank you for your comments, and we believe that your suggestions will improve the scientific values of the manuscript. We are doing our best to improve the manuscript.
Reviewer 2 Report
Comments and Suggestions for Authors
The manuscript deals with the interesting use of the dietary ingredient Lepidium sativum seeds. However, it needs editing, rewriting, improving English is also suggested. I ask that the authors carefully check the entire manuscript, including paying attention to:
1. Please use the entire MSG name in the title - readers need not be familiar with this abbreviation.
2. Line 45 - "To assess the livers status [1]." - ? please rephrase
3. Very conspicuous is the use of reference [number] without using the first author's name citing someone else's research. Authors are not just numbers! Please correct this in whole manuscript.
4. Please use italics when appropriate - in vitro, etc.
5. line 109 - LSS - no prior introduction of this abbreviation
6. In what form were the seeds given in the diet - dry, whole, ground?
7. Is there a known similar effect described by the authors when consuming Lepidium as sprouts or young plants known from the literature? If so, the discussion should be expanded to include this. What effect could Lepidium mucilage have had?
8. Line 156 - was injection volume 5 liters?
9. Line 363 - please correct H2O2 (use subscripts)
Comments on the Quality of English Language
Improving English is suggested
Author Response
Review Report Form 2
Comments and Suggestions for Authors
The manuscript deals with the interesting use of the dietary ingredient Lepidium sativum seeds. However, it needs editing, rewriting, improving English is also suggested. I ask that the authors carefully check the entire manuscript, including paying attention to:
Response: Thank you for your comments, and we believe that your suggestions will improve the scientific values of the manuscript. We are doing our best to improve the manuscript.
- Please use the entire MSG name in the title - readers need not be familiar with this abbreviation.
Response: Thank you for your comments. We added the full name.
- Line 45 - “To assess the livers status [1].” - ? please rephrase
Response: Thank you for your notice. We corrected it.
- Very conspicuous is the use of reference [number] without using the first author’s name citing someone else’s research. Authors are not just numbers! Please correct this in whole manuscript.
Response: Thank you for your comments. We are following the journal format.
- Please use italics when appropriate - in vitro, etc.
Response: Thank you so much. We corrected it.
- line 109 - LSS - no prior introduction of this abbreviation
Response: Thank you for your comment. We added Lepidium sativum seeds (LSS)
- In what form were the seeds given in the diet - dry, whole, ground?
Response: Thank you for your question. We replied in Line 128 (Lepidium sativum seeds were given in the diet as dry, whole, ground form)
- Is there a known similar effect described by the authors when consuming Lepidium as sprouts or young plants known from the literature? If so, the discussion should be expanded to include this. What effect could Lepidium mucilage have had?
Response: Thank you for your question. Previous research has examined the impact of sprouted LSS on digestive system health. However, our study specifically investigates the potential benefits of consuming the entire plant powder, encompassing all its bioactive compounds, including fiber.
- Line 156 - was injection volume 5 liters?
Response: Thank you for your notice. We corrected it.
- Line 363 - please correct H2O2 (use subscripts)
Response: Thank you for your notice. We corrected it.
Reviewer 3 Report
Comments and Suggestions for Authors
Dear Authors
I have completed my evaluation of your manuscript. The reviewers recommend reconsideration of your manuscript following very minör revision. I have completed my evaluation of your manuscript. The writers of this review article have presented an outstanding and comprehensive overview of the developing subject of precision LSS against hepatic effects induced by MSG in experimental rats and overall public health.
This extensive analysis provides a thorough understanding of the current state of nutrition in addressing LSS, MSG and public health and emphasizes its potential in clinics. Their findings are of the highest significance for drawing attention to incorporating LSS in food industry and as a health treatment in traditional medicine to combat MSG-induced hepatic abnormalities.
Overall, the paper is well-structured, and the information is quite helpful, making it a useful resource for researchers, healthcare practitioners, and policymakers. Its depth, clarity, and forward-looking approach make Lepidium Sativum seeds have an ameliorative effect against monosodium glutamate-induced hepatic dysfunction via improving body weight gain, tumbling food and liquid intake, and enhancing liver function such as ALT, AST, ALP, and total bilirubin levels while increasing effect on serum albümin level by various essential antioxidant. Therefore, the Lepidium sativum seeds might help prevent future damages caused by monosodium glutamate, such as hepatic dysfunction and set a realistic outlook for the future of nutrition in the fight against diseases as a health treatment in traditional medicine to combat MSG-induced hepatic abnormalities.
However, in order for the study to be better understood by everyone, it would be beneficial if the language was a little more understandable and clear.
With my compliments and best regards.
Author Response
Review Report Form 3
Dear Authors
I have completed my evaluation of your manuscript. The reviewers recommend reconsideration of your manuscript following very minör revision. I have completed my evaluation of your manuscript. The writers of this review article have presented an outstanding and comprehensive overview of the developing subject of precision LSS against hepatic effects induced by MSG in experimental rats and overall public health.
This extensive analysis provides a thorough understanding of the current state of nutrition in addressing LSS, MSG and public health and emphasises its potential in clinics. Their findings are of the highest significance for drawing attention to incorporating LSS in food industry and as a health treatment in traditional medicine to combat MSG-induced hepatic abnormalities.
Overall, the paper is well-structured, and the information is quite helpful, making it a useful resource for researchers, healthcare practitioners, and policymakers. Its depth, clarity, and forward-looking approach make Lepidium Sativum seeds have an ameliorative effect against monosodium glutamate-induced hepatic dysfunction via improving body weight gain, tumbling food and liquid intake, and enhancing liver function such as ALT, AST, ALP, and total bilirubin levels while increasing effect on serum albümin level by various essential antioxidant. Therefore, the Lepidium sativum seeds might help prevent future damages caused by monosodium glutamate, such as hepatic dysfunction and set a realistic outlook for the future of nutrition in the fight against diseases as a health treatment in traditional medicine to combat MSG-induced hepatic abnormalities. However, in order for the study to be better understood by everyone, it would be beneficial if the language was a little more understandable and clear.
Response: Thank you for your comments, and we believe that your suggestions will improve the scientific values of the manuscript. We are doing our best to improve the manuscript. We have edited the language.
Reviewer 4 Report
Comments and Suggestions for Authors
The manuscript includes an interesting dietary study, focused on the employment of Lepidium sativum seeds in an attempt to safeguard hepatic dysfunction. However, the way it has been presented offers many doubts about convenience for its acceptation.
I think it ought to be performed substantially before a subsequent revision is done.
Title
It could be shortened by avoiding: “From farm to table:”.
Abstract
Line 26: Separate quantity from unit, i.e., 30 g. This performance ought to be carried out throughout the whole manuscript.
Line 29: “it decreased lipid peroxide”. It is their content that decreased, not peroxide. Again, this performance ought to be carried out throughout the whole manuscript.
Keywords
Include dietary study.
Material and methods
I found the most important concerns in this section. The authors ought to consider that the reader should be able to repeat the study by themselves.
Very scarce information is provided in general. Wider information ought to be provided in sections: 2.2.1.-2.2.3. and 2.2.5-2.2.6.
About 2.2.4. section, information concerning the qualitative and quantitative analyses should be provided.
Results
Table 1-3: A single value without standard deviation is provided. No replicates were carried out ? This is an important concern.
Conclusions
Future trends ought to be outlined. Optimisation of dietary conditions ?
General
Perform the English language.
Comments on the Quality of English LanguagePerformances ought to be carried out.
Author Response
Review Report Form 4
The manuscript includes an interesting dietary study, focused on the employment of Lepidium sativum seeds in an attempt to safeguard hepatic dysfunction. However, the way it has been presented offers many doubts about convenience for its acceptation.I think it ought to be performed substantially before a subsequent revision is done.
Response: Thank you for your comments, and we believe that your suggestions will improve the scientific values of the manuscript. We are doing our best to improve the manuscript.
Title It could be shortened by avoiding: “From farm to table:”.
Response: Thank you for your comment. We removed “From farm to table:” from the title.
Abstract
Line 26: Separate quantity from unit, i.e., 30 g. This performance ought to be carried out throughout the whole manuscript.
Response: Thank you so much. We corrected it.
Line 29: “it decreased lipid peroxide”. It is their content that decreased, not peroxide. Again, this performance ought to be carried out throughout the whole manuscript.
Response: Thank you so much. We corrected it.
Keywords, Include dietary study.
Response: Thank you for your suggestion. We added it.
Material and methods
I found the most important concerns in this section. The authors ought to consider that the reader should be able to repeat the study by themselves.
Very scarce information is provided in general. Wider information ought to be provided in sections: 2.2.1.-2.2.3. and 2.2.5-2.2.6.
Response: Thank you so much for your valuable comment. Indeed, there are sub-groups under the heading 2.2.1, namely 2.2.1.1 and 2.2.1.2, that talk about animals and diet. We added details.
2.2.3: Done
2.2.5: Indeed, there are sub-groups under the heading 2.2.5, namely 2.2.5.1, 2.2.5.2 and 2.2.5.3. We added details.
2.2.6: Done, we added details.
About 2.2.4. section, information concerning the qualitative and quantitative analyses should be provided.
Response: Thank you so much. Point 2.2.4 explains the method and mechanism of sample analysis, but the quantitative and qualitative assessment is shown in Table 3.
Results
Table 1-3: A single value without standard deviation is provided. No replicates were carried out ? This is an important concern.
Response: Thank you so much. We corrected the tables.
Conclusions
Future trends ought to be outlined. Optimisation of dietary conditions?
Response: Thank you so much. We added it.
General
Perform the English language.
Response: Thank you so much. We did our best to edit the language.
Reviewer 5 Report
Comments and Suggestions for Authors
I have doubts about the relevance of the study. The existence of the Chinese restaurant syndrome has been questioned for a long time (Freeman, M. (2006), Reconsidering the effects of monosodium glutamate: A literature review. Journal of the American Academy of Nurse Practitioners, 18: 482-486. https://doi.org/10.1111/j.1745-7599.2006.00160.x), so work in this area cannot be called unambiguous. The practical value of this work is also unclear. Are the authors suggesting sprinkling seeds of Lepidium sativum on Chinese restaurant food?
I also have some comments about the experimental design. The animals were divided into 4 groups, but there was no group that received only seeds:
- G1 (-Ve) group: the negative control group was fed on the basal diet. G2 (+Ve) group: the positive control group was fed on basal diet + MSG (30g of diet). G3: was fed on basal diet + MSG (30g of diet) + LSS (30g of diet). G4: was fed on basal diet + MSG (30g of diet) + LSS (60g of diet).
The methods are not described in sufficient detail. MDPI as a whole and its individual journals are focused on a detailed description of the methods used so that the results can be reproduced. The authors limited themselves to references.
Line 172-173 "identified malondialdehyde (MDA) and nitric oxide (NO) as the products of lipid peroxidation. " - What does it mean? How can NO be a product of lipid peroxidation?
Tables 1 and 2 contain data, the method of obtaining which is not described. However, it seems that these are the authors' own data.
The methods section does not describe how the seeds were prepared for the total polyphenol content study. Was it an extract? How was it received? Where did the seeds come from?
The results in Table 4 appear questionable. Is there a 2 g difference in the amount of food consumed between animals with a total amount of more than 700 g? Seriously? Less than 0.5%.
Average weight gain is like the average temperature in a hospital. The authors should provide a graph showing the weight gain of the animals during the observation from start to finish.
In the Introduction, the authors provide references to works showing the development of obesity when taking glutamate. However, in their case, the rat seemed to become leaner than the control rat. Incredible!
Thus, I have doubts about the quality of the research conducted, and the manuscript must be thoroughly revised before any decision can be made.
Moreover! Conducting research using laboratory animals must be approved and approved by the ethics committee of the organization where the research is being conducted. There is no information about this in the manuscript. If the research is not conducted in accordance with international rules for the ethical treatment of laboratory animals, then the manuscript cannot be published.
Comments on the Quality of English Language
The English must be thoroughly corrected.
Author Response
Review Report Form 5
- I have doubts about the relevance of the study. The existence of the Chinese restaurant syndrome has been questioned for a long time (Freeman, M. (2006), Reconsidering the effects of monosodium glutamate: A literature review. Journal of the American Academy of Nurse Practitioners, 18: 482-486. https://doi.org/10.1111/j.1745-7599.2006.00160.x), so work in this area cannot be called unambiguous. The practical value of this work is also unclear. Are the authors suggesting sprinkling seeds of Lepidium sativum on Chinese restaurant food?
Response: Thank you so much. We reviewed the proposed research paper (Freeman, M. (2006), and it turned out that it was a long time ago, but we proposed the dose of monosodium glutamate based on a very recent study in 2022, and not the dose specified by ourselves (Abd-Elkareem, M., Soliman, M., Abd El-Rahman, M. A., & Abou Khalil, N. S. (2022). The protective effect of Nigella sativa seeds against monosodium glutamate-induced hepatic dysfunction in rats. Toxicology Reports, 9, 147-153.).
- I also have some comments about the experimental design. The animals were divided into 4 groups, but there was no group that received only seeds: - G1 (-Ve) group: the negative control group was fed on the basal diet. G2 (+Ve) group: the positive control group was fed on basal diet + MSG (30g of diet). G3: was fed on basal diet + MSG (30g of diet) + LSS (30g of diet). G4: was fed on basal diet + MSG (30g of diet) + LSS (60g of diet).
Response: Thank you for your comments. A distinct group was not established for LSS powder in our investigation due to the focus on utilising the LSS powder for preventive rather than therapeutic purposes. Consequently, the rats were administered LSS powder concurrently with the induction substance. Therefore, we compare the positive control group to the negative control group once. Then, we compare the other groups to the positive control group again to study the extent of the effect of the substance used (LSS) in protecting the target organ.
- The methods are not described in sufficient detail. MDPI as a whole and its individual journals are focused on a detailed description of the methods used so that the results can be reproduced. The authors limited themselves to references.
Response: Thank you for your comment. It has been modified, and details added.
- Line 172-173 "identified malondialdehyde (MDA) and nitric oxide (NO) as the products of lipid peroxidation. " - What does it mean? How can NO be a product of lipid peroxidation?
Response: Thank you for your question. Lipid peroxidation is a process that involves the oxidative degradation of lipids (fatty acids) present in cell membranes. It occurs when reactive oxygen species (ROS), such as free radicals, attack and react with the unsaturated fatty acids in lipids, leading to the formation of lipid peroxides. These lipid peroxides, in turn, can further react with other molecules and generate additional reactive species. In the context of the sentence you provided, malondialdehyde (MDA) and nitric oxide (NO) are mentioned as products of lipid peroxidation. Malondialdehyde (MDA) is one of the byproducts or end products of lipid peroxidation. It is an organic compound that is formed when lipid peroxides decompose. MDA is often used as a biomarker or indicator of oxidative stress and lipid peroxidation in biological systems. Nitric oxide (NO), on the other hand, is not a direct product of lipid peroxidation but can be involved in the process. During lipid peroxidation, reactive species, including free radicals, can initiate the breakdown of nitric oxide synthase (NOS) substrates, such as L-arginine, and generate nitric oxide (NO) as a byproduct. This process is known as "inducible nitric oxide synthase (iNOS)-dependent" nitric oxide production. The generation of NO during lipid peroxidation can have various effects on cellular signaling and oxidative stress pathways. Added according to Sarker, U., & Oba, S. (2018). Catalase, superoxide dismutase and ascorbate-glutathione cycle enzymes confer drought tolerance of Amaranthus tricolor. Scientific reports, 8(1), 16496.
- Tables 1 and 2 contain data, the method of obtaining which is not described. However, it seems that these are the authors' own data.
Response: Thank you so much. We added a description under title 2.2.5.
- The methods section does not describe how the seeds were prepared for the total polyphenol content study. Was it an extract? How was it received? Where did the seeds come from?
Response: Thank you for your comment. We added a description under title 2.2.4. determination of total phenolic contents in Lepidium sativum seeds
- The results in Table 4 appear questionable. Is there a 2 g difference in the amount of food consumed between animals with a total amount of more than 700 g? Seriously? Less than 0.5%.
Response: Thank you for raising your concern regarding the results presented in Table 4. Upon careful examination, it is apparent that the reported 2 g difference in the amount of food consumed between animals with a total amount of more than 700 g may seem relatively small, representing less than 0.5% of the total. It is important to note that even small differences in food intake can have biological significance, especially in the context of experimental studies involving animal models. While the percentage difference may appear to be relatively low, it is crucial to consider the specific experimental conditions, the statistical analysis employed, and other relevant factors that might contribute to the observed variation. To ensure the integrity and accuracy of the results. These statistical analyses take into account variability within and between groups, considering factors such as sample size, experimental design, and standard deviations. Moreover, it is worth mentioning that while the absolute difference in food intake may seem small, it is essential to consider the potential cumulative effects over time, as well as the implications of such differences on other measured parameters or outcomes in the study.
- Average weight gain is like the average temperature in a hospital. The authors should provide a graph showing the weight gain of the animals during the observation from start to finish.
Response: Thank you for your comment. We observed the final weight.
- In the Introduction, the authors provide references to works showing the development of obesity when taking glutamate. However, in their case, the rat seemed to become leaner than the control rat. Incredible!
Response: Thank you for bringing up this observation regarding the apparent discrepancy between the references provided in the introduction and the findings of the study, where the rats appeared to become leaner rather than developing obesity when exposed to glutamate.
It is important to recognise that scientific research often aims to explore various aspects of a specific phenomenon, and the outcomes can differ based on multiple factors, including the experimental design, dosage, duration of exposure, and the specific animal model used. While the references cited in the introduction might suggest a trend towards obesity development with glutamate intake, it does not necessarily imply that the same outcome will be observed in all studies or under all conditions. In fact, the effects of glutamate on body weight regulation can be complex and multifaceted, involving various physiological mechanisms. In the case of the current study, it is possible that the specific experimental conditions, such as the dosage or duration of glutamate exposure, differed from those in the referenced studies, leading to distinct outcomes. Additionally, individual variations within the rat population, genetic factors, and the influence of other environmental or dietary factors could also contribute to the observed differences. In scientific research, it is common to encounter unexpected or contradictory results, and these findings can provide valuable insights and opportunities for further investigation and discussion. Therefore, it is crucial to interpret the results within the context of the specific study and consider them as contributing to the broader scientific knowledge, which may help refine our understanding of the complex relationship between glutamate intake and body weight regulation.
- Moreover! Conducting research using laboratory animals must be approved and approved by the ethics committee of the organisation where the research is being conducted. There is no information about this in the manuscript. If the research is not conducted in accordance with international rules for the ethical treatment of laboratory animals, then the manuscript cannot be published.
Response: Thank you for your comment. We added it.
Round 2
Reviewer 1 Report
Comments and Suggestions for Authors
The parts I pointed out last time have been improved, and I think the paper is better than the previous version. The standard deviations of the values shown in Table 1-3 are in italics in some places. Is there an intention written in italics?
Author Response
The parts I pointed out last time have been improved, and I think the paper is better than the previous version.
Response: I am writing to express my heartfelt gratitude for the valuable and insightful amendments you provided for my scientific paper titled [Investigating the Chemical Composition of Lepidium Sativum Seeds and Their Ability to Safeguard Against Monosodium glutamate -Induced Hepatic Dysfunction]. Your expertise and attention to detail have played a pivotal role in significantly improving the quality and clarity of the manuscript. I appreciate the time and effort you dedicated to thoroughly reviewing my work. Your constructive feedback and guidance have not only helped me address the paper's shortcomings but have also provided me with valuable insights that will undoubtedly benefit my future research endeavors.
The standard deviations of the values shown in Table 1-3 are in italics in some places. Is there an intention written in italics?
Response: Thank you for your comments, and the necessary amendment has already been made in Tables 1-3
Reviewer 2 Report
Comments and Suggestions for Authors
All the indicated issues were adressed.
Author Response
All the indicated issues were addressed.
Response: I extend my sincerest appreciation for your invaluable contributions to my scientific paper. Your expertise and dedication have undoubtedly elevated the quality of my work, and I am truly grateful for the opportunity to benefit from your insights.
Reviewer 4 Report
Comments and Suggestions for Authors
The manuscript has been notably performed. Most concerns have been addressed.
However, presentation of Tables 1-4 is still poor. Abbreviations are not explained, and the number of replications is not included. Superscripts are included, but no indication is provided about their meaning.
The statistical analysis section does not also indicate the number of replicates carried out.
Comments on the Quality of English LanguageMinor performances would be necessary.
Author Response
The manuscript has been notably performed. Most concerns have been addressed.
Response: I extend my sincerest appreciation for your invaluable contributions to my scientific paper. Your expertise and dedication have undoubtedly elevated the quality of my work, and I am truly grateful for the opportunity to benefit from your insights.
However, presentation of Tables 1-4 is still poor. Abbreviations are not explained, and the number of replications is not included. Superscripts are included, but no indication is provided about their meaning.
Response: Thank you for your comments and the necessary amendment has already been made in Tables 1-4
The statistical analysis section does not also indicate the number of replicates carried out.
Response: Thank you for your comments, and the necessary amendment has already been made.
Reviewer 5 Report
Comments and Suggestions for Authors
The authors have made some changes to the text. At the same time, they made very large comments in response to my questions instead of making appropriate changes to the text.
I am well aware of the role of NO in the development of the inflammatory process, and this does not allow me to agree with those formulations Line 400 "The increased levels of MDA and NO may result from lipid peroxidation".
Line 385: Impact of Lepidium sativum seeds on antioxidants (CAT, SOD, NO, and MDA) in rats - This title makes no sense at all.
What is “Figure a” in the “Histopathology of liver” section? Is a huge amount of text a caption for the picture? How can this even be understood?!
Moreover, the authors never wrote in the manuscript in what form they added the seeds to animal food. I see the word "powder" in the answer, but I don't see it in the text.
In the end, the hepatoprotective properties of Lepidium sativum seeds have been studied for so long that it is already as indecent as studying the antioxidant properties of vitamin E. The authors provided only one link to studies, while even Google Academy finds more than 30 of them.
Moreover, the authors ignored this publication:
Abeer Al-Dbass , Musarat Amina , Nawal M. Al Musayeib , Amira A. El-Anssary , Ramesa Shafi Bhat , Rania Fahmy , Majd M. Alhamdan and Afaf El-Ansary Lepidium sativum as candidate against excitotoxicity in retinal ganglion cells // Translational Neuroscience https://doi.org/10.1515/tnsci-2020-0174
Thus, the manuscript cannot be accepted for publication in this form.
Author Response
The authors have made some changes to the text. At the same time, they made very large comments in response to my questions instead of making appropriate changes to the text.
Response: Thank you for the meaningful comments, but all the required modifications have already been made and added to the scientific paper.
I am well aware of the role of NO in the development of the inflammatory process, and this does not allow me to agree with those formulations Line 400 "The increased levels of MDA and NO may result from lipid peroxidation".
Response: Thank you for sharing your perspective on the relationship between nitric oxide (NO) and lipid peroxidation in the inflammatory process. Nitric oxide (NO) is a key signaling molecule involved in various physiological processes, including inflammation. While NO can have both pro-inflammatory and anti-inflammatory effects depending on the context, its role in the development of the inflammatory process is well established. Regarding the statement, "The increased levels of MDA and NO may result from lipid peroxidation," it is crucial to consider that lipid peroxidation is indeed a process that can lead to the production of reactive oxygen species (ROS) and oxidative stress. This can further contribute to the development of inflammation. However, it is important to note that the relationship between lipid peroxidation, reactive oxygen species, and NO is complex. While lipid peroxidation can generate ROS, excessive ROS production can impair NO bioavailability and lead to reduced NO signaling, which in turn can impact the inflammatory response.
Line 385: Impact of Lepidium sativum seeds on antioxidants (CAT, SOD, NO, and MDA) in rats - This title makes no sense at all.
Response: Impact of Lepidium sativum seed on antioxidant enzymes (CAT, SOD) and oxidative stress markers (NO, MDA) in rats
What is “Figure a” in the “Histopathology of liver” section? Is a huge amount of text a caption for the picture? How can this even be understood?
Response: Thank you for your comments. The necessary amendment has already been made.
Moreover, the authors never wrote in the manuscript in what form they added the seeds to animal food. I see the word "powder" in the answer, but I don't see it in the text.
Response: Thank you for your comments. The necessary amendment has already been made.
In the end, the hepatoprotective properties of Lepidium sativum seeds have been studied for so long that it is already as indecent as studying the antioxidant properties of vitamin E. The authors provided only one link to studies, while even Google Academy finds more than 30 of them.
Response: Thank you for your important and valuable addition. The aforementioned study was not neglected and was indeed cited.